# Method for Detecting Disorder of a Nonlinear Dynamic Plant

**DOI:** 10.3390/s25041256

**Published:** 2025-02-19

**Authors:** Xuechun Wang, Vladimir Eliseev

**Affiliations:** 1Department of Control and Intelligent Technologies, National Research University “Moscow Power Engineering Institute”, Krasnokazarmennaya 14, Moscow 111250, Russia; vlad-eliseev@mail.ru; 2InfoTeCS JSC, Mishina 56, Moscow 127083, Russia

**Keywords:** disorder detection, time series, dynamic plant, cross-correlation function, neural network autoencoder, cumulative sum algorithm, exponentially weighted moving variance, pH neutralization reaction

## Abstract

This paper proposes a new disorder detection method CCF-AE for a scalar dynamic plant based only on its input–output relation using a cross-correlation function and neural network autoencoder. The CCF-AE method does not use the reference model of the dynamic object, but only considers real-time behavior changes, given by input and output time series. The proposed method was used to detect disorder in the process of a nonlinear pH neutralization reaction, and was compared with the cumulative sum control chart (CUSUM) and the exponentially weighted moving variance control chart (EWMV). The CCF-AE method demonstrates a better true detection rate and lower false alarm rate than CUSUM and EWMV. Also, CCF-AE has more advantages in detecting disorder of complex nonlinear processes.

## 1. Introduction

In increasingly complex industrial environments, the stable operation of automation control systems is crucial to ensuring improved production efficiency and product quality. With the development of information technology, traditional control system design has gradually shifted from analog circuit architectures that rely on continuous signal processing to discrete time control technology using intelligent algorithms. However, this change also brings with it new challenges—the increasing complexity of systems and the diversity of failure modes [1]. The complexity of systems highlights the problem of detecting disorder and anomalies. Disorder is an unpredictable change in system parameters and an anomaly is a visible effect of system behavior change.

Disorder detection plays a vital role in modern automated control systems. It means dynamic monitoring for the rapid detection of any behavior or pattern deviating from the regular operating state during system operation by analyzing real-time data, which is a so-called abnormal situation. The root of abnormal situations includes but is not limited to hardware performance degradation, operation errors, and communication interruptions, which may cause system instability or even crashes.

The use of disorder detection technology helps to provide an early warning, enables the taking of preventive measures to prevent potential problems from worsening, and helps to maintain the efficient operation and safety of the entire system.

There are many situations that can lead to anomalies in the behavior of dynamic plants. Anomalies can occur individually or simultaneously due to defects in sensors, actuators, or plants. However, in complex industrial systems, it is sometimes not easy to accurately monitor whether a certain performance status as a system health indicator is normal (i.e., whether it is a fault or not); for example, the degree of machine wear and changes in material properties. These subtle changes may not be obvious at first, and we realize the problem only when they reach a certain level and cause system performance to degrade or fail. When monitoring multiple process parameters simultaneously, skilled operators often must make operating decisions based solely on their experience. Another reason for disorder in the behavior of dynamic plants can be changes in external influences on the system. This may be a change in both external conditions leading to a change in the dynamic characteristics of the system components and the setpoint signal being different from expected.

In the maintenance of complex systems, fault detection faces severe challenges. First of all, real-time monitoring is crucial. The system must react and handle exceptions almost instantly and is sensitive to delays. Secondly, the data sent by the sensor may contain errors, such as inaccurate measurements or random fluctuations caused by changes in the external environment, which requires the algorithm to be highly reliable and resistant to interference. Furthermore, the continuous updating and adaptability of the fault detection rules is also a challenge. Such rules are usually based on some reference model of normal system behavior and threshold settings. However, such rules (the model and thresholds) sometimes need to updated due to a possible slow shift in normal behavior. The update procedure should not require maintenance by a serviceman or an engineer, otherwise it would be too expensive and not practical.

Modern approaches to disorder detection combine machine learning [2] and big data analytics, and may incorporate cloud computing and IoT technology to enable data analysis using information not available in traditional control systems. From a practical perspective, the challenge for such approaches is to process large amounts of data efficiently in real time, while identifying disorder and ensuring the reliability of the detection algorithms so that they can cope with changing conditions and new causes of abnormal behavior in dynamic plants.

## 2. Background

The detection of a disorder in the behavior of dynamic plants has been an active research area for many years [3,4].

Algorithms for detecting disorder are usually based on real-time data. If the deviation between the plant output and the output of its reference model increases, an alarm or automatic adjustment mechanism is triggered. The simplest approach was to monitor the range of variation in the observed characteristics of the plant [5]. More complex approaches, such as the cumulative sum algorithm (CUSUM), are based on detecting the deviation in the statistical characteristics of the observed process from the normal behavior specified by the stochastic model of the norm. Typically, such a model implies the specification of the type and parameters of the distribution of the normal process.

In recent years, thanks to the significant enhancements in computer performance, disorder detection techniques using machine learning have gradually emerged, such as autoencoders [6,7]. As a powerful tool, the autoencoder can autonomously learn the basic structure and distribution of data and then reconstruct them. So, the data samples of normal working conditions will be mapped to similar ones, while the abnormal samples will not, resulting in a large difference between input and output vectors of the autoencoder. Therefore, these points that deviate from the norm will be effectively identified. This not only improves the accuracy and reliability of anomaly detection, but also reduces the need for manual intervention, which helps to build a more intelligent and adaptive industrial monitoring system.

### 2.1. Model-Based Methods

Model-based disorder detection is a widely used technology in science and engineering. Its basic principle is to provide a mathematical description or algorithm for the behavior of the main characteristics of a system. This requires the use of mathematical formalisms to analyze the key physical principles and observed dependencies, often by constructing differential equations to characterize the dynamic processes. Model-based analysis is particularly effective when designing systems with clear structures and well-defined rules, such as simple mechanical devices in mechanical engineering or linear control systems in the field of automatic control, because models are easy to create and debug.

The task of detecting faults in systems and processes is to automatically identify dependencies between measured signals and subsequently detect changes in these dependencies. Based on the measured output signals of the plant and its reference model, modeling error values, estimated values of parameters, or estimated values of state variable values called features can be calculated. Changes or deviations in the detected features compared to normal features will lead to the analysis of symptoms [8,9].

A typical reference model-based disorder detection scheme is shown in Figure 1.

This approach is based on the assumption that when the physical characteristics within the plant are slightly adjusted, the measured output signal will change accordingly. If this change deviates from the standard signal pattern estimated by the reference model, it will become obvious and can be identified as an abnormal situation. When using this approach, it is necessary to take into account that the reference model must accurately repeat the behavior of the plant itself and similarly respond to input effects on the plant.

For simple processes, model-based methods give good results. However, when dealing with complex dynamic systems, such as chemical reaction processes or systems involving a large number of variables, building an accurate reference model is often difficult or even nearly impossible. In such complex situations, detection methods that rely on building models of dynamic plants may not be applicable. Instead, a data-driven approach that does not require building accurate models of dynamic plants may be considered. Data-driven approaches are better suited for complex systems with high uncertainty because of their greater emphasis on analyzing empirical data rather than building detailed analytical mathematical models of the plant.

### 2.2. Data-Driven Methods

Data-driven anomaly detection, as a modern anomaly diagnosis strategy, is based on a large amount of operational data as a primary resource. Its main idea is to collect data generated by the equipment under normal operating conditions, such as temperature, vibration, current, and other measured parameters, and then use these time series to build a model. Data-based models can capture the evolution of plant performance over time and use it as the basis for distinguishing whether the plant is in normal condition.

The specific process includes data collection, pre-processing (cleaning, normalization, etc.), feature engineering (extracting key features that cause failures), and then training a model to define the normal operating mode of the plant.

When new data are fed into the model, if they significantly deviate from the known data distribution under normal operating conditions, an alarm will be triggered, indicating a potential failure. Data-driven methods such as machine learning can automatically learn patterns from observed historical data, better capture nonlinear correlations, and avoid relying on precise physical models, making them superior in dealing with nonlinear system problems.

Data-driven disorder detection technology is now widely used in various industrial processes including the chemical industry [10], polymer manufacturing, microelectronics, the steel industry, pharmaceutical processes, power distribution networks [11], and flow systems.

Especially in the last three decades, it has become one of the most fruitful areas of research and practice and an important tool for quality control [12,13].

#### 2.2.1. Multivariate Statistical Process Control

Multivariate statistical process control in process monitoring emerged in the 1990s [14]. It uses statistical models to monitor manufacturing processes in real time by extracting key information from complex datasets. For example, statistical process control charts are used to compare the current state of a process to normal operating conditions. Most commonly, control charts are used to monitor the mean or deviation of selected variables that affect a process; these include the Shewhart control chart [15], the CUSUM chart [16], and the EWMA chart [17]. Each of the above types of control charts has its own advantages and disadvantages. The Shewhart control chart is a statistical tool based on past data fluctuations. It is designed with the assumption that samples are collected and analyzed continuously. When focusing only on existing test data, if improvements, anomalies, or slight changes in the process occur in a short period of time, the control chart may not immediately reflect these short-term changes due to the slow sample update, because it relies on historical means and standard deviations to determine the normal range. In short, control charts have a good ability to identify stable long-term trends, but they may not be sensitive enough for dynamic and fast-changing processes, especially those with frequent small adjustments. In contrast, the CUSUM and EWMA control charts are more sensitive in detecting small changes in the process because they use information from long-sequence samples. Therefore, for large-scale process failures that require an immediate response, the Shewhart control chart is an effective tool, but for continuous improvement or monitoring quality trends, other methods such as EWMA or CUSUM control charts may be needed to monitor the process in more detail. In practical applications, they are often used together to detect anomalies and evaluate process stability [18].

In general, multivariate statistical methods for disorder detection use the typical Hotelling’s $T^{2}$ statistic and the Q statistic, which is also known as the squared prediction error. When the $T^{2}$ value exceeds a certain threshold, it means that the data points deviate from the normal pattern, i.e., deviations from the normal plane may be observed and, thus, disorder is detected.

In the field of process control, principal component analysis (PCA) is one of the most popular statistical methods for extracting information from measured data [19]. The main step of PCA is to transform the original data into a low-dimensional linear independent space and residual space by linear projection. $T^{2}$ and Q statistics and their control limits are set in the corresponding space for fault detection.

However, when dealing with complex processes in fields such as industrial chemistry and biology, the use of PCA can be problematic if these processes involve significant nonlinear features such as periodic changes or adaptive responses within the system. Since the basic assumption of using PCA is that data changes are normally distributed and linearly related, its ability to detect nonlinear outliers will be weakened. For example, when outliers are generated by nonlinear relationships or the behavior of the data evolves over time rather than following a simple linear trend, a simple linear transformation of PCA may not accurately capture this outlier behavior of the system. To overcome this shortcoming, several improved PCA schemes have been proposed. For instance, a previous paper [20] presented a nonlinear principal component analysis based on a five-layer neural network.

Partial least squares (PLS) is also a classic multivariate statistical analysis technique, and is used to establish a linear relationship between the input and output. By building a model based on normal data in an a priori order, PLS can be implemented in forecasting and monitoring applications. Assuming that all measurements follow a normal distribution, $T^{2}$ and the Q index are adopted as statistical indicators for detecting quality and non-quality defects [21]. In contrast to PCA, independent component analysis has been used for process monitoring by projecting correlated variables into an independent space without orthogonality constraints, making it more applicable to non-Gaussian processes [22]. Another method for dealing with non-Gaussian processes is the Gaussian mixture model, which fits multiple Gaussian models to approximate non-Gaussian data. It has been widely used for process monitoring in arbitrary datasets that do not follow a normal distribution [23,24].

#### 2.2.2. Combination of Reference Model and Control Chart Approach

Model-based methods may be combined with control charts to detect disorder as a systematic difference between the plant and reference model behavior (Figure 1). Assuming a stochastic input signal, the difference between plant output and the expected behavior predicted by the reference model may be observed as an increase in the variance of the error $d=y-\hat{y}$, representing the discrepancy between the measured plant output y and the modeled output $\hat{y}$.

Let us collect a set of errors $d_{1},d_{2},\ldots ,d_{k}$, initially assumed to follow a probability density function $\rho (d,\sigma_{0}^{2})$. When a disorder occurs at some unknown moment, these errors would then follow a different distribution characterized by another probability density function $\rho (d,\sigma_{1}^{2})$.

When running a one-sided CUSUM control chart to identify process disorders, the statistics $S_{k}$ is calculated as:(1)$$S_{k}=\left\{\begin{array}{c} 0,k=0\\ \max\left(0;S_{k-1}+z_{k}\right),k>0 \end{array}\right.$$
where the term $z_{k}$ denotes a statistic used to detect shifts or changes in the process variance and is defined as follows:(2)$$z_{k}=-\frac{1}{2}ln\frac{\sigma_{1}^{2}}{\sigma_{0}^{2}}-\frac{1}{2}\left(\frac{1}{\sigma_{1}^{2}}-\frac{1}{\sigma_{0}^{2}}\right)d_{k}^{2}$$

When the cumulative sum exceeds the control limit H, the process is considered out of control and the disorder is detected.(3)$$S_{k}>H$$

In this case, the elementary verification procedure concludes. If further analysis is required, the CUSUM control chart can be reapplied to the sequence $z_{k}^{j}$, where j denotes the iteration index.

The one-sided exponentially weighted moving variance control chart (EWMV) is also used to detect process disorder. The EWMV statistic at the current time k is:(4)$$Q_{k}=\left(1-r\right)Q_{k-1}+r{\left(d_{k}-\mu_{k-1}\right)}^{2}$$
where r∈(0, 1] is a weighting factor that governs the rate of exponential discounting of the past data, $Q_{0}$ represents the initial estimate of the mean squared error, and $\mu_{k}$ is the estimation of the process mean at time k by exponentially weighted moving average (EWMA), calculated by:(5)$$\mu_{k}=\left(1-\lambda \right)\mu_{k-1}+\lambda d_{k}$$
where λ∈(0, 1] is a smoothing coefficient.

The current EWMV statistic should also be compared to a predefined control limit H. Specifically, if $Q_{k}>H$, it indicates potential disorder. The determination of the control limit (threshold) H for control charts generally depends on empirical experience and involves balancing the true detection rate with the false alarm rate.

#### 2.2.3. Machine Learning-Based Approach

Machine learning models can automatically extract features and learn complex patterns from large amounts of data without manually developing complex hypotheses or hypothesis distributions. This is especially effective when dealing with nonlinear and non-Gaussian distributed data. Some machine learning methods, such as support vector machines or artificial neural networks, have a certain noise resistance and the ability to handle isolated points. In recent years, there has been a growing interest in the application of artificial neural networks in fault detection and diagnostic systems [25,26,27].

The neural network replaces the analytical model describing the process under normal operating conditions. It must be trained to perform this task, and the training data can be collected directly from the process or from a simulation model that is as realistic as possible. Once the training is complete, the neural network can generate residuals that indicate anomalies. For example, a previous paper [28] described using the residuals between the output of a plant and its neural network model to detect anomalies in a real sugar evaporation process.

However, it is necessary to consider dynamics when modeling dynamic plants in automatic control systems using neural networks or detecting disorder in the behavior of a plant. In order to be able to record the dynamic behavior of the system, the neural network must have dynamic characteristics, such as recurrent neural networks. A previous paper [29] discusses a method for online detection of sensor disorder in substations based on a long-term short-term memory network and an adaptive threshold selection algorithm.

Using only a single fault detection algorithm may lead to false positive problems. In order to improve the accuracy and stability of detection, a common approach is to integrate multiple technical means, such as combining statistical analysis and machine learning methods. Such a comprehensive strategy can complement the advantages of each method, reduce the possibility of errors, and enhance the credibility of the final result.

A previous paper [30] proposed a process monitoring method using Gaussian mixture model and weighted kernel-independent component analysis, in which the Gaussian mixture model is used to estimate the probability of kernel-independent components, and the important kernel-independent components that dominate the process change are given a higher weight according to the estimated probability of collecting important information in the online disorder detection process.

Another paper [31] adopted a method using automatic noise accumulation coding and a K nearest neighbor (KNN) rule. An autoencoder with multi-level noise reduction is used to model nonlinear process data and automatically extract important features. The original nonlinear space is then mapped to the feature space and residual space using the multi-level denoising autoencoder. Two new statistics for detecting disorder in the above spaces are constructed by introducing a KNN rule with corresponding control limits determined by the kernel density estimation.

Although machine learning has shown great potential in the field of disorder detection, it still faces some challenges. In an industrial environment, continuous evolutionary processes may occur, such as equipment aging and changing failure modes. It can be difficult for traditional static anomaly detection methods to account for these changes. In addition, anomalies are often not a single pattern, but contain noise and unknown variables. If a model relies too heavily on a known data distribution, it may fail when it encounters new anomalies that were not previously noticed.

The main idea of this work is the use of the cross-correlation function (CCF) between the input and output of the plant as a characteristic of its nonlinear dynamics, combined with a neural network autoencoder, which helps detect when the plant operation shifts to a new pattern and when anomalies occur compared to the reference period of system operation. The proposed method does not rely on external reference signals, but directly analyzes real-time data from nonlinear dynamic systems, which can respond more quickly to real-time changes within the monitored plant. In addition, the autoencoder can dynamically learn the normal distribution of the data during the training process, thereby avoiding the subjectivity and variability caused by manual threshold setting.

This idea was developed by the authors earlier and was applied to tasks of operating mode change detection [32], disorder detection for linear ARMA and nonlinear AR dynamic processes [33], and quality control of an autonomous neural network dynamic model [34]. In this study, we extend the application of the CCF and neural network autoencoder approach to solving the problem of detecting disorder of a nonlinear dynamic object defined by a mathematical model.

For comparison, we conducted experiments using the proposed method, the CUSUM control chart, and the EWMV control chart to detect disorder in the behavior of the same plant.

## 3. Method

### 3.1. Cross-Correlation Functions as Characteristics of a Dynamic System

The cross-correlation function (CCF) is a statistical measure that quantifies the similarity between two signals as they change over time. In dynamic systems, any change in the operational state—whether caused by internal hardware variations or external disturbances—can impact the CCF between the input and output signals. Such impacts often appear as shifts in correlation peaks, variations in amplitude, or irregular waveform distortions. Therefore, by regularly monitoring the CCF between inputs and outputs, one can observe and remember how the dynamic system behaves under different operating conditions [34].

For each successive time step *t* of the control system, the CCF between the input u(t) and the corresponding output y(t) is given as follows:(6)$$r_{uy}\left(\tau \right)=\int_{-\infty }^{\infty }u\left(t\right)y\left(t+\tau \right)dt$$
where τ∈R is a shift distance variable between the input and the output.

The discrete sampling of continuous signals *u*(*t*) and *y*(*t*) is denoted as *u*(*n*) and *y*(*n*), respectively (where t=nΔt,n∈Z). Over a finite time interval of width *d*, the cross-correlation function at the k-th time interval of the input and output time series is calculated as:(7)$$r_{uy}\left(\tau ,k\right)=\frac{1}{d}\sum_{n=k}^{k+d-1}u\left(n\right)y\left(n+\tau \right)$$
where τ∈Z represents the argument of the CCF. The CCF of the k-th time interval can be represented as a vector $R_{uy}^{k}={\left[{r_{uy}\left(-d+1,k\right),\ldots ,r}_{uy}\left(0,k\right),\;\ldots ,r_{uy}\left(d-1,k\right)\right]}^{T}\in R^{2d-1}$.

The set of CCF vectors describes the behavior of the dynamic system in response to the input signal over a specified time period. During normal system operation, the CCF vectors calculated at various time intervals form a point cloud within a specific spatial region. Figure 2 shows an example of such a region in three-dimensional space, denoted as $R_{uy}^{N}$, where the superscript N means “normal”, reflecting typical operational conditions.

When the input deviates from the normal input within the $R_{uy}^{N}$ region, the values in the CCF vector will also change accordingly. Assume that the system’s dynamic response at time p is represented by the point $R_{uy}^{p}$, which belongs to the normal region $R_{uy}^{N}$. Consequently, if the dynamic response changes at some later moment (q>p), the new CCF will be denoted by the point $R_{uy}^{q}$, as shown in Figure 2.

A drift $R_{uy}^{p}\to R_{uy}^{q}$ detected in the CCF vector space outside the region $R_{uy}^{N}$ representing the normal behavior of the system means that at time q, either a change in the input signal or a modification in the properties of the dynamic system has occurred. We call this phenomenon disorder.

### 3.2. Disorder Detection Using Neural Network Autoencoder

In order to use the cross-correlation function (CCF) as a characteristic representation for detecting disorders in a dynamic system, two problems need to be solved. One is how to effectively store and retain the CCF calculated during normal system operation as a reference pattern, and the other is how to quantify the discrepancies between the current calculated cross-correlation function and their corresponding reference patterns.

For this, we used a neural network autoencoder capable of encoding and reconstructing the CCF vectors. The autoencoder (AE) learns the representation of data in an unsupervised manner by utilizing vectors ($R_{uy}^{k}$, k∈[1,M]) calculated from the input and output sequences during an interval of the system’s normal operation. The autoencoder aims to map each sample within this dataset into a hidden space, with the objective that this hidden space encoding can accurately reconstruct the original CCF vector.

The autoencoder primarily comprises an encoder and a decoder, as shown in Figure 3. The encoder $z=f\left(W_{enc}x+b_{enc}\right)$ compresses the set of CCF vectors into a compact, low-dimensional representation known as the code vector. The decoder $\tilde{x}=f(W_{dec}z+b_{dec})$ then aims to reconstruct the original CCF vectors from this compressed representation by applying the inverse transformation:(8)$$\tilde{x}=AE\left(x\right)=f(W_{dec}z+b_{dec})$$
where x is the vector CCF, $W_{enc}$, $W_{dec}$ are weight matrices, and $b_{enc}$, $b_{dec}$ are bias terms for the encoder and decoder, respectively.

In our study, the learning objective of the autoencoder is to minimize the loss *E*. After training, the autoencoder acquires the ability to model the distribution patterns inherent in the CCF vectors computed during normal operation periods of a dynamic system.(9)$$E=\sum_{k=1}^{M}{\left\|x^{k}{-\tilde{x}}^{k}\right\|}^{2}$$
where $x^{k}$ is the vector of the k-th CCF vector and ${\tilde{x}}^{k}$ is its reconstruction, and $\left\|z\right\|=\sqrt{\sum_{k=1}^{M}z_{k}^{2}}$ is usual Euclidean length.

When a new CCF vector is fed into the autoencoder, if the output of the decoder closely approximates to the input, this indicates that the system functions normally within the corresponding time interval as represented by the CCF vector. Conversely, if the reconstructed data from the autoencoder differs significantly from the input—specifically when the reconstruction error r exceeds a predefined threshold $r_{th}$—it suggests disorders in the system during that period.(10)$$r(k)=\left\|x^{k}{-\tilde{x}}^{k}\right\|$$

### 3.3. Development and Application of a Disorder Detector

In this section, we detail the development process and application methods of the disorder detector.

Figure 4 shows the scheme for synthesizing an autoencoder-based disorder detector, and its steps are as follows:

*Step 1*: Data preparation. Let us collect a discrete time series of length L from both the input $u\left(l\right),1\leq l\leq L$ and the output $y\left(l\right)$ generated by a normally functioning dynamic system, and select L to be proportional to d, which represents the width of the cross-correlation function (CCF) calculation window. Also, let us split the entire time series $u\left(l\right)$ and $y\left(l\right)$ into segments of length d. Specifically, for each segment (or window) indexed by *k*:

Input vector: $u^{k}={\left[u(\left(k-1\right)d+1),\ldots ,u\left(kd\right)\right]}^{T}\in R^{d},1\leq k\leq \frac{L}{d}$Output vector: $y^{k}={\left[u(\left(k-1\right)d+1,\ldots ,y\left(kd\right)\right]}^{T}\in R^{d},1\leq k\leq \frac{L}{d}$.

*Step 2*: Calculation of CCF. Let us calculate CCF using Equation (7) with the vectors $u^{k}$ and $y^{k}$ and obtain a sequence of CCF vectors $R_{uy}^{k}$ describing the reference (normal) behavior of the dynamic system.

*Step 3*: Training of an autoencoder. The training dataset D of the autoencoder is a set of CCF vectors $x^{k}=R_{uy}^{k}$, which characterize the normal behavior of the dynamic system. Let us train the autoencoder on dataset D with the objective of minimizing the loss function *E* through error backpropagation or a similar optimization technique, aiming to obtain the optimal reconstructed ${\tilde{x}}^{k}=AE\left(x^{k}\right)$. The trained autoencoder provides a smaller error for CCF vectors for those closer to the original and a larger error for those that deviate more significantly.

*Step 4*: Determination of a threshold. To detect the deviations of CCF from its normal state, it is necessary to setup a threshold for the reconstruction error. Let us define the trigger threshold $r_{th}$ as the maximum of the reconstruction error rth=max k⁡rk obtained during training. If the reconstruction error exceeds $r_{th}$, it indicates that the CCF deviates significantly from the expected pattern.

For convenience, we call the synthetized detector CCF-AE and apply it to detect whether the system g(⋅) has dynamic disorder. The detection scheme is shown in Figure 5 and consists of the following 4 steps:

*Step 1*: Collect a time series of consecutive input values $u\left(\left(k-1\right)d+1\right),\ldots ,u\left(kd\right)$ and output values $y\left(\left(k-1\right)d+1\right),\ldots ,y\left(kd\right)$ at the time sample kd, where d is the width of the window to calculate CCF and k is an index of the window. Let us arrange them into vectors $u^{k}={\left[u\left(\left(k-1\right)d\right),\ldots ,u\left(kd\right)\right]}^{T}\in R^{d}$ and $y^{k}={\left[y\left(\left(k-1\right)d\right),\ldots ,y\left(kd\right)\right]}^{T}\in R^{d}$.

*Step 2*: Calculate the CCF using Equation (7) with vectors $u^{k}$ and $y^{k}$, obtaining the vector $x^{k}=R_{uy}^{k}$.

*Step 3*: Calculate the reconstructed CCF $\tilde{x}(k)$ using Equation (8). Then, substitute the results of Equations (7) and (8) into Equation (10) to calculate the reconstruction error r(k).

*Step 4*: Compare the reconstruction error r(k) with the threshold $r_{th}$. If $r(k)\leq r_{th}$, it indicates that the dynamic characteristics of the system are still within a controllable range, consistent with the behavior patterns previously learned and memorized by the autoencoder, Otherwise, it suggests that there is disorder in the system g(⋅) in the kth time interval of duration *d*.

In summary, the dynamic characteristics represented by the CCF vector $R_{uy}^{k}$ are used as input data for the autoencoder to obtain the reconstruction error for assessing the behavior of the dynamic system, which allows identifying and localizing disturbances in the dynamic system.

Let us evaluate the proposed method for a real-world nonlinear dynamic plant—a mathematical model of the pH neutralization reaction process.

## 4. Case Study: pH Neutralization Process

### 4.1. Mathematical Model of a pH Neutralization Reactor

Consider a one-dimensional pH neutralization reactor in a control system, where complex mixing processes between acidic and alkaline liquids occur. These processes vary significantly over time and under different conditions, involving a series of nonlinear chemical reactions that lead to complex and potentially unpredictable system dynamics. The pH neutralization process described by Hall and Seborg (1989) was used as a simulation platform [35], as illustrated in Figure 6. The inputs to the neutralization reactor include the base stream ($q_{1}$), acid stream ($q_{3}$), and buffer stream ($q_{2}$). The measured output (*y*) from the neutralization reactor is the pH of the effluent solution. The controller regulates the flow of alkali into the tank by adjusting the opening degree of the valve based on the specified target pH value, thereby controlling the reagent mixing rate via the control signal $u=q_{1}$. The buffer stream ($q_{2}$), acid stream ($q_{3}$), and tank volume are assumed to remain constant during this process.

The dynamic model for the reaction invariants of the outgoing solution ${(W}_{a},W_{b})$ in state-space form is given by the following expression:(11)$$\dot{x}\left(t\right)=f_{1}\left(x\left(t\right)\right)u\left(t\right)+f_{2}\left(x\left(t\right)\right)d\left(t\right)+f_{3}\left(x\left(t\right)\right)$$(12)$$F\left(x,y\right)=0$$(13)$$x≜{\left[x_{1},x_{2}\right]}^{T}={\left[W_{a},W_{b}\right]}^{T}$$(14)$$f_{1}\left(x\left(t\right)\right)={\left[\frac{1}{V}\left(W_{a_{1}}-x_{1}\right),\frac{1}{V}\left(W_{b_{1}}-x_{2}\right)\right]}^{T}$$(15)$$f_{2}\left(x\left(t\right)\right)={\left[\frac{1}{V}\left(W_{a_{2}}-x_{1}\right),\frac{1}{V}\left(W_{b_{2}}-x_{2}\right)\right]}^{T}$$(16)$$f_{3}\left(x\left(t\right)\right)={\left[\frac{q_{3}}{V}\left(W_{a_{3}}-x_{1}\right),\frac{q_{3}}{V}\left(W_{b_{3}}-x_{2}\right)\right]}^{T}$$(17)$$F\left(x,y\right)=x_{1}\left(t\right)+10^{y\left(t\right)-14}-10^{-y\left(t\right)}+x_{2}(t)\frac{1+2\times 10^{y\left(t\right)-K_{2}}}{1+10^{K_{1}-y(t)}+10^{y\left(t\right)-K_{2}}}$$

Equations (11) and (12) were solved using the Runge–Kutta method 45. The state variables $W_{a}$ and $W_{b}$ are the response invariants. Model parameters are provided in Table 1, while the nominal operating point is detailed in Table 2.

### 4.2. Simulation Modeling of the pH Neutralization Reactor

To simulate the behavior of the pH neutralization, the model proposed by [35] was implemented in the SimInTech (version 2.24.1.26) software [36], which is an analog to MATLAB/Simulink 2024a. Figure 7 shows the schema of the model simulation.

To obtain the process behavior in a given range of operating conditions, an amplitude-modulated pseudo-random signal (APRBS) was used as an input control signal. Real-world noise was simulated by adding Gaussian white noise to the output, with a signal-to-noise ratio of 60. The number of sampling steps was 26. The disturbance was simulated by the additional sinusoidal input signal with an amplitude of 2 and a frequency of $\frac{0.2}{2\pi }$ in a time interval from the 3001st to the 4154th sample. The process input was confined to the range u∈[12.5,17], while the output pH value was restricted to the range y∈[6,9].

An example of the input and output time series of the pH neutralization reactor produced by the model (Figure 7) is represented in Figure 8.

### 4.3. Methodology of Comparative Experiments

#### 4.3.1. Competing Methods

To demonstrate the effectiveness of the proposed CCF-AE method in action and to compare its performance with existing approaches, several experiments were performed. For a competition with the proposed CCF-AE method, two well-known disorder detection methods were considered: the cumulative sum control chart (CUSUM) for the variance of the error and the exponentially weighted moving variance control chart (EWMV). Their algorithms are represented on Figure 9.

#### 4.3.2. Quality Indicators

The determination of the control limit (threshold) H often involves a trade-off between the false alarm rate $R_{FAR}$ and true detection rate $R_{TDR}$. Let us consider them as metrics to evaluate the performance of a disorder detection method:(18)$$R_{FAR}=\frac{FP}{FP+TN}$$(19)$$R_{TDR}=\frac{TP}{TP+FN}$$
where TP—data sample correctly labeled as a disordered point, FN—disordered points that are mistakenly labeled as normal, FP—normal data that are incorrectly labeled as disordered, TN—data correctly labeled as normal.

$R_{FAR}$ is the probability that the system issues an alarm but no disorder actually occurs. $R_{TDR}$ is the probability that the system successfully triggers an alarm when a disorder occurs. A higher threshold will reduce $R_{FAR}$, but may also reduce $R_{TDR}$. A lower threshold will increase $R_{TDR}$ but may cause higher false alarms.

In addition, the choice of threshold also affects the time of average alarm’s delay $T_{ad}$ and the mean time between false alarms $T_{fa}$.

#### 4.3.3. CUSUM Control Chart Implementation

To implement a CUSUM control chart for disorder detection, sufficient samples *d* must be collected to estimate $\sigma_{0}^{2}$ and $\sigma_{1}^{2}$. Additionally, an appropriate threshold H should be determined to ensure that the quality indicators meet acceptable standards.

The steps to run the CUSUM control chart are as follows:

*Step 1*: Collect data. Gather the output error $d_{k}$ between the plant and its reference model.

*Step 2*: Initialize cumulative sum. Initialize the cumulative sum statistics $S_{0}=0$.

*Step 3*: Calculate $S_{k}$. Calculate the $z_{k}$ at each moment and add it to the sum of all previous $S_{k-1}$.

*Step 4*: Determine whether the threshold has been crossed. When the cumulative sum exceeds the threshold H, it is considered a disorder.

*Step 5*: Monitoring and alarming. Keep monitoring the changes in the cumulative sums. Once the $S_{k}$ exceeds the threshold, an alarm is triggered and that point is recorded as the start time of the disorder. Then reset the cumulative sum back to zero and the entire process is restarted to continue monitoring the process.

#### 4.3.4. EWMV Control Chart Implementation

The steps to run the EWMV control chart are as follows:

*Step 1*: Collect data. Gather the output error $d_{k}$ between the plant and its reference model.

*Step 2*: Initialize EWMV. Initialize the EWMV $Q_{0}=\sigma_{0}^{2}$. Initialize coefficients *r* and λ.

*Step 3*: Calculate $Q_{k}$. Calculate ${\left(d_{k}-\mu_{k}\right)}^{2}$ at each moment and add it to the sum of all previous $Q_{k-1}$ after multiplying it by weight *r*.

*Step 4*: Determine whether the threshold has been crossed. When $Q_{k}$ exceeds the threshold *H*, it is considered a disorder.

*Step 5*: Monitoring and alarming. Keep monitoring the changes in the cumulative sums. Once $Q_{k}$ exceeds the threshold, an alarm is triggered and that point is recorded as the start time of the disorder. Then reset $Q_{k}$ back to zero and the entire process is restarted to continue monitoring the process.

## 5. Case Study Experiments

### 5.1. The Reference Model Synthesis and Evaluation

To apply the control chart approach for detecting disorder in a dynamic system, it is necessary to first create its reference model. Given the nonlinear characteristics of the pH neutralization reactor, we used the GRU neural network.

The GRU network has an input u(n), a layer with nine GRU units, a fully connected layer with 15 neurons, and a regression layer to construct the output $\hat{y}(n)$. The total length of the training dataset is 10,000. The training lasted 400 epochs. The result of using GRU to model the neutralization reaction is RMSE=0.25.

Figure 10 shows both the actual plant output y(n) and the output ${\hat{y}}_{GRU}(n)$ from the GRU network. Figure 11 shows the error $d(n)=y(n)-{\hat{y}}_{GRU}(n)$, which is useful for applying control charts.

### 5.2. Detection Using CCF-AE Algorithm

We calculate $R_{uy}^{k}$ with a step size of *d* = 5, and this vector represents the characteristics of the system within a time interval of length *d*.

It is appropriate to choose a time interval (window) length of 5. Increasing the window length will increase the detection time. A shorter time series corresponds to resolution loss at lower frequencies. If some important characteristics of the system are reflected in the low-frequency part, shortening the window may not enable fully obtaining the low-frequency features.

Figure 12 shows the CCF vectors $R_{uy}^{k}$ under both normal operating conditions and during periods of disorder. Different colors in the figure represent CCF values across various time windows. It can be seen that during disorder, the shape and amplitude of the CCF vector undergo significant changes that are enough to be detected by autoencoder.

To develop the CCF-AE detector, we collected a dataset $D={\left\{u\left(n\right),y(n)\right\}}_{n=1}^{5000}$ from the normal operation process to calculate $R_{uy}^{k}$, where k∈[1, 1000]. Each vector $R_{uy}^{k}$ was normalized and used as part of the training set for the autoencoder. The autoencoder employs a multilayer perceptron neural network architecture with 10 neurons in the hidden layer. For each time instance, the input vector $R_{uy}^{k}\in R^{2d-1}$ is fed into the autoencoder, aiming to reconstruct itself at the output. Training proceeded over 300 epochs. The maximum reconstruction error observed during the training process was established as a threshold.

Next, we used the trained CCF-AE detector to detect disorder in the process. The disorder occurred at samples from 3001 to 4154. Similarly, we collected $u^{k}\in R^{d}$ and $y^{k}\in R^{d}$ at every *d*-time interval to calculate $R_{uy}^{k}\in R^{2d-1}$; the reconstruction error *r* between the reconstructed vector calculated by the autoencoder and the original $R_{uy}^{k}$ exceeds the threshold in the 601st to 831st windows. The result converted to the time axis is shown in Figure 13.

Figure 13 shows that the reconstruction error *r* significantly exceeds the threshold (red dashed line) between the 3005th and 4155th samples, closely aligning with the start and end times of the disorder. Detection results yielded true positives (*TPs*) = 980, false positives (*FPs*) = 0, and false negatives (*FNs*) = 174. The calculated quality metrics are summarized in Table 3, demonstrating that the CCF-AE detector has no false alarms and has a high $R_{TDR}$. The entire detection process required 0.4006 s. The experiments were conducted on a computer with the following specifications: CPU—Intel Core i7-13620H; RAM—16 Gb; SSD—1024 Gb; graphics card—NVIDIA GeForce RTX 4050 Laptop GPU; Operating System—Windows 11. The device manufacturer, Lenovo, is located in Jinan, China.

### 5.3. Detection Using CUSUM Control Chart

For the initialization of the CUSUM control chart, the variance determined for the mode before disorder $\sigma_{0}^{2}=0.0060$ should be taken, and the variance during disorder $\sigma_{1}^{2}=3\sigma_{0}^{2}$ should be adopted to represent the expected increase in variability. The threshold *H* = 100 was empirically determined through experimentation with historical data.

Figure 14 shows the disorder detection results using the CUSUM control chart. *TP* = 167, *FP* = 22, and *FN* = 987 were detected. The calculated quality indicators are represented in Table 3. The entire detection process, including the calculation of the output of the reference model, took 0.2584 s.

### 5.4. Detection Using EWMV Control Chart

The reference model was established from prior experiments. For the exponentially weighted moving variance (EWMV) control chart, the initial parameters were derived based on pre-disorder data: variance $\sigma_{0}^{2}=0.0048$ and mean $\mu_{0}=-0.1649$. The threshold *H* = 0.0008 and the coefficients *r* = 0.7, λ=0.8 were determined through empirical analysis of historical data.

Figure 15 shows the disorder detection results using the EWMV control chart. A total of 658 true positives (TPs), 20 false positives (FPs), and 496 false negatives (FNs) were identified. The calculated quality metrics are summarized in Table 3. The entire detection process, including the calculation of the output of the reference model, required 0.2623 s.

### 5.5. The Comparison of CCF-AE, CUSUM and EWMV

Table 3 presents the quality metrics for the disorder detection results obtained using all three methods.

By comparing the results of all three competing methods, it can be concluded that the CCF-AE detector exhibits a higher true detection rate and a lower false alarm rate.

## 6. Discussion

In this study, we propose using the input–output cross-correlation function (CCF) as a dynamic characteristic indicator for system behavior, and combining it with the reconstruction error of the autoencoder to perform disorder detection in the process.

A significant advantage of the CCF-AE method in disorder detection is its reliance on a single input and output signal from the system, thereby eliminating the necessity to account for additional process variables.

Compared with traditional single-variable process monitoring techniques such as CUSUM and EWMV, CCF-AE has the following advantages:Calculating the cross-correlation function and training the autoencoder are less complex compared to constructing a sufficiently accurate plant dynamic model and adjusting the parameters of the control chart.CCF-AE is more adaptable to nonlinear systems due to directly estimating quasi-linear dynamics within the current time interval. In contrast, statistical process control methods require an accurate system model, which may demand significant computational resources.

The proposed method also has certain limitations:A fixed detection delay of CCF-AE may increase the false alarm and missed alarm rates under highly stringent real-time requirements.Compared with statistical process control methods, CCF-AE incurs higher computational costs, although it is well suited for modern digital microcontrollers equipped with hardware-implemented floating-point operations.

We emphasize the importance of optimizing key method parameters, specifically the cross-correlation function (CCF) window width, autoencoder neural network architecture, and error threshold. These parameters significantly influence system performance as follows:CCF Window Width: A narrower CCF window width enhances detection sensitivity by capturing more short-term dynamic changes. Conversely, a larger window width helps smooth out signals, reducing noise interference, although at the cost of increased response latency. An appropriate window width strikes a balance between sensitivity and stability.Autoencoder Neural Network Architecture: The number of layers, the number of nodes per layer, and the choice of activation functions in the autoencoder directly affect the model’s learning capability and generalization ability. More complex architectures improve feature extraction but increase the risk of overfitting, diminishing the adaptability to unseen data. Striking a balance between simplicity and sufficient expressive power is essential to ensure robustness without unnecessary complexity.Error Threshold: Setting the threshold too high leads to missed detections, while setting it too low causes frequent false alarms or misclassifications. Optimizing this threshold involves integrating insights from historical data analysis, expert domain knowledge, and specific user requirements.

In summary, the optimization of these parameters ensures that the system achieves optimal performance across various operating conditions.

## 7. Conclusions

Considering the limitations of traditional disorder detection methods that require a reference model and some a priori knowledge about the statistical properties of signals in the system, a disorder detection method CCF-AE based on the cross-correlation function of the input–output response of the plant and the neural network autoencoder is established.

Compared with CUSUM and EWMV control charts, the CCF-AE detector demonstrated a higher true detection rate in experimental evaluations. Additionally, CCF-AE did not generate any false alarms during the experiments, resulting in both a false alarm rate and mean time between false alarms of 0. In contrast, these quality metrics for the CUSUM and EWMV control charts were significantly greater than zero.

In addition, one key advantage of the CCF-AE method is its independence from a reference model. It solely relies on the input and output signals of the system, making it suitable for detecting disorders in nonlinear systems without involving the internal mechanisms of the plant.

While the computational complexity of the CCF-AE detector is higher than that of CUSUM and EWMV control charts, these control charts require an additional plant prediction model, which adds substantial computational overhead for training and application.

This study demonstrates the effectiveness of the CCF-AE detection technology through a representative case involving a pH neutralization reactor. This conclusion is based on a single application, which may not fully represent its performance in other scenarios. However, we have based this research on previously conducted works [33,34,35], where we solved other similar tasks using the same approach with CCF and AE. Notably, the design of the CCF-AE detector does not depend on specific nonlinear plant dynamics models, indicating its potential generality and adaptability across various settings. Nevertheless, we acknowledge the necessity of validating its performance using more diverse datasets to comprehensively evaluate both strengths and limitations.

To further improve the quality and scope of application of the research results, two primary improvements are planned to be taken: one is to expand the scale of real-world datasets for testing; the other is to introduce more diverse data-driven technologies.

## Figures and Tables

**Figure 1 sensors-25-01256-f001:**
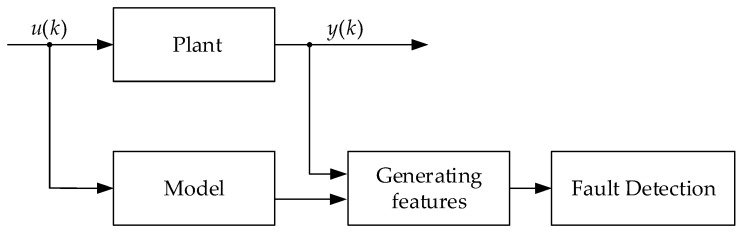
Scheme for the reference model-based disorder detection.

**Figure 2 sensors-25-01256-f002:**
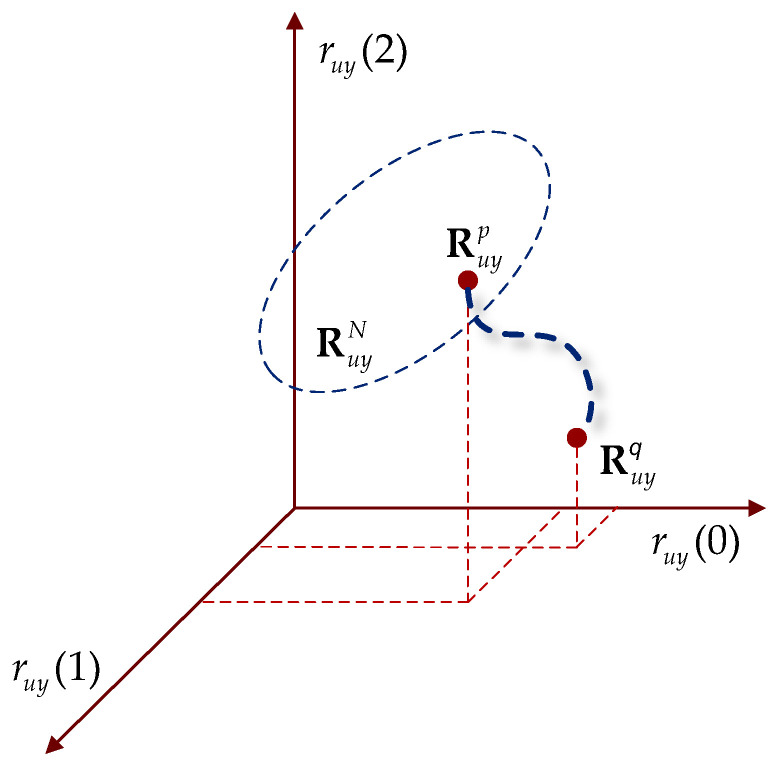
Vector space of cross-correlation functions.

**Figure 3 sensors-25-01256-f003:**
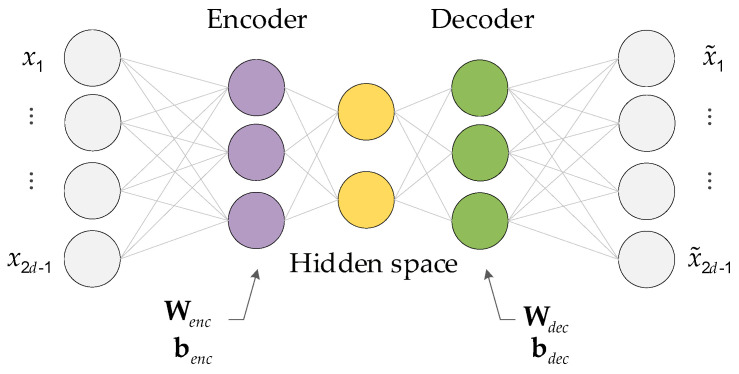
Architecture of the neural network autoencoder.

**Figure 4 sensors-25-01256-f004:**
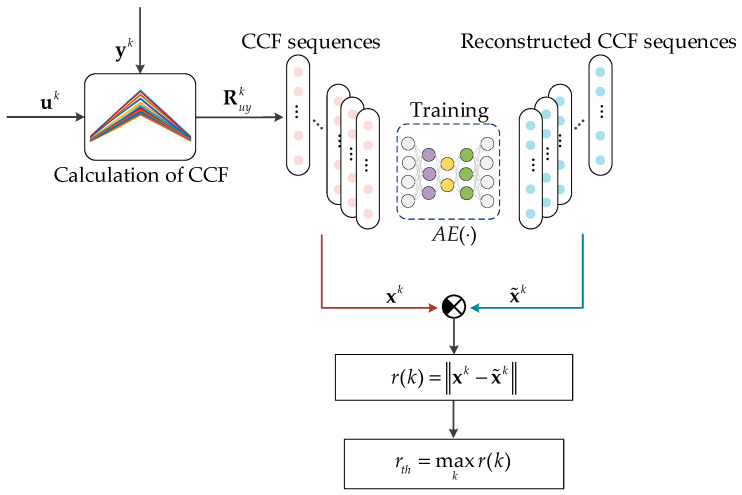
Scheme for the synthesis of disorder detector.

**Figure 5 sensors-25-01256-f005:**
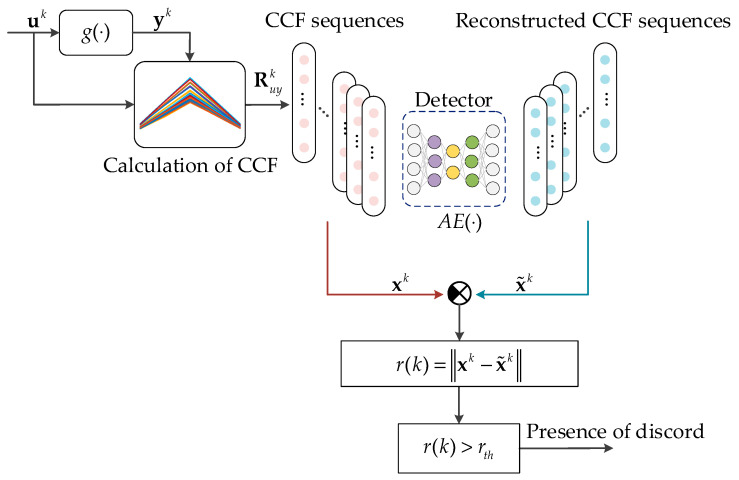
Scheme for the application of disorder detector.

**Figure 6 sensors-25-01256-f006:**
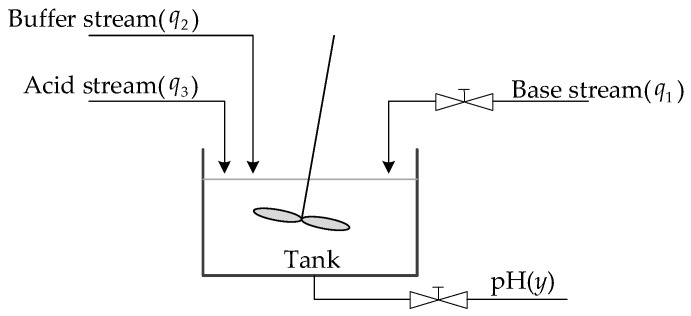
The process of the pH neutralization reaction.

**Figure 7 sensors-25-01256-f007:**
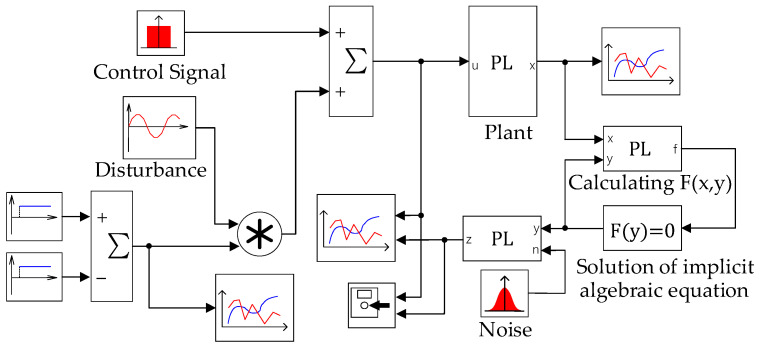
Screenshot of the SimInTech program window with a simulation model of the pH neutralization reaction.

**Figure 8 sensors-25-01256-f008:**
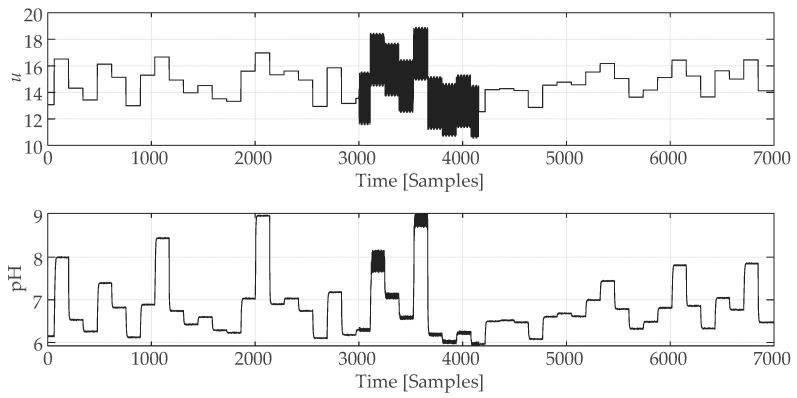
Results of the neutralization reactor simulation.

**Figure 9 sensors-25-01256-f009:**
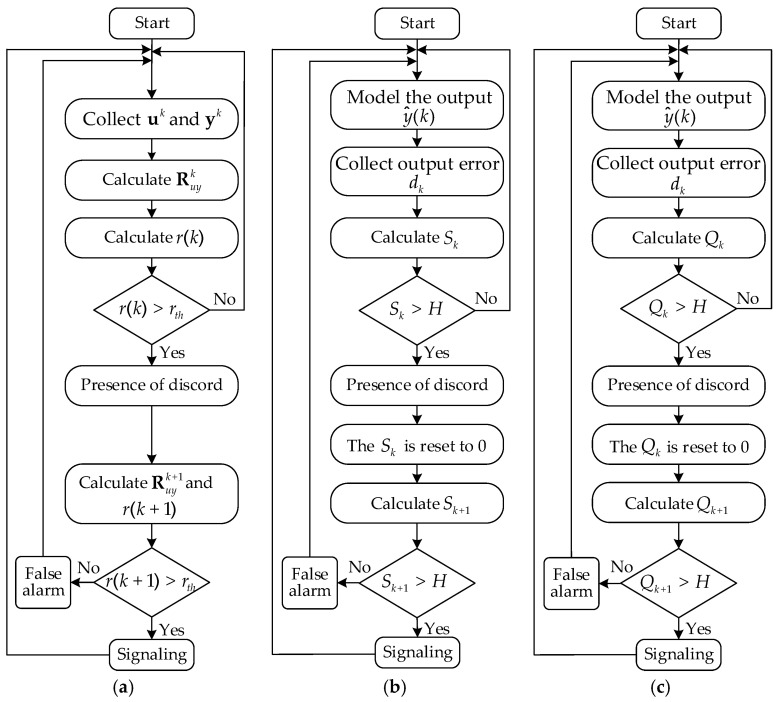
Algorithms for detecting disorders: CCF−AE (**a**), CUSUM (**b**), EWMV (**c**).

**Figure 10 sensors-25-01256-f010:**
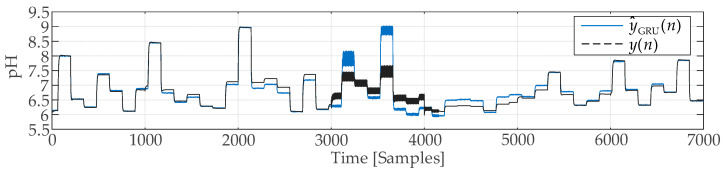
Simulation of neutralization reaction using GRU neural network.

**Figure 11 sensors-25-01256-f011:**
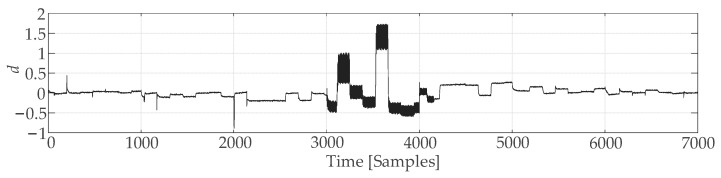
Identification error of GRU neural network.

**Figure 12 sensors-25-01256-f012:**
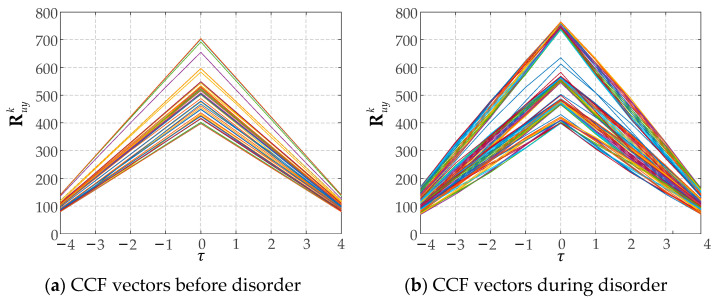
CCF vectors in different windows.

**Figure 13 sensors-25-01256-f013:**
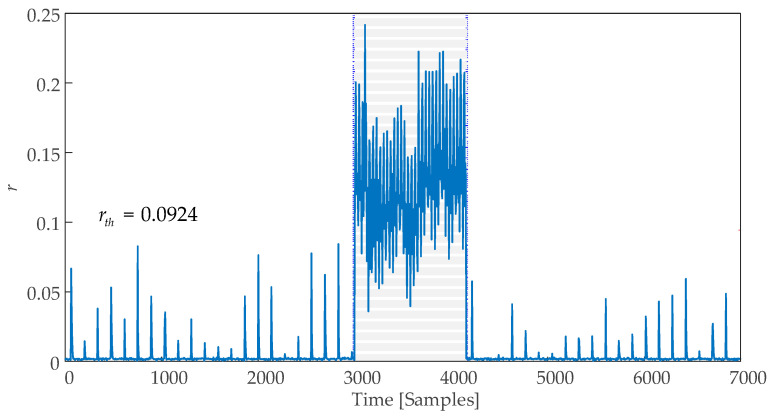
Detection trajectory of CCF-AE.

**Figure 14 sensors-25-01256-f014:**
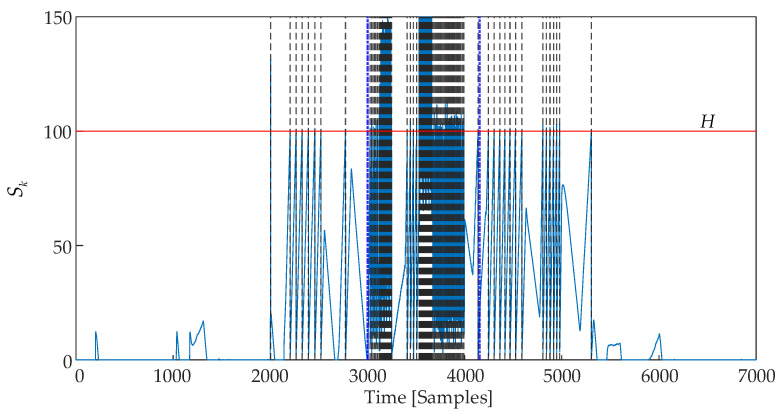
CUSUM control chart.

**Figure 15 sensors-25-01256-f015:**
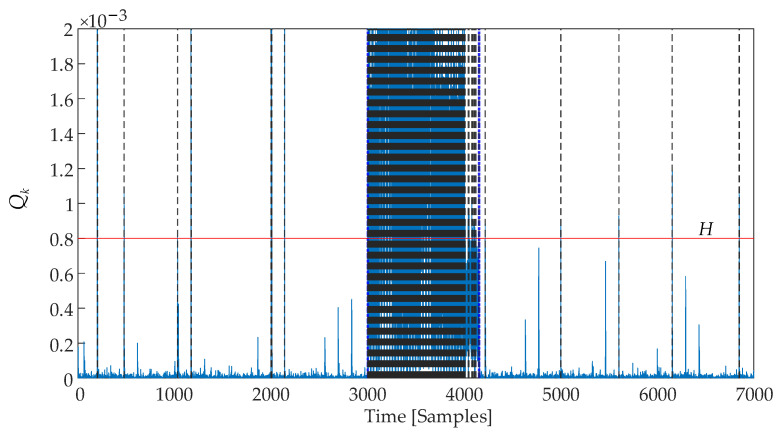
EWMV control chart.

**Table 1 sensors-25-01256-t001:** Parameters for the baseline model of the neutralization reactor.

Parameter	Value
$W_{a_{1}}$	$-3.05\times 10^{-3}\;mol$
$W_{a_{2}}$	$-3\times 10^{-2}\;mol$
$W_{a_{3}}$	$3\times 10^{-3}\;mol$
$W_{b_{1}}$	$5\times 10^{-5}\;mol$
$W_{b_{2}}$	$3\times 10^{-2}mol$
$W_{b_{3}}$	0
V	2900 mL
$K_{1}$	6.35
$K_{2}$	10.25

**Table 2 sensors-25-01256-t002:** Nominal operating points for the neutralization reactor.

Parameter	Value
$W_{a}$	$-4.32\times 10^{-4}\;mol$
$W_{b}$	$5.28\times 10^{-4}mol$
$q_{1}$	15.55 mL/s
$q_{2}$	0.55 mL/s
$q_{3}$	16.60 mL/s

**Table 3 sensors-25-01256-t003:** Quality and computational cost indicators for disorder detection.

Algorithm	$R_{FAR}$	$R_{TDR}$	$T_{ad}$	$T_{fa}$	Calculation Time (s)
CCF-AE	0	0.8492	5.8673	0	0.4006
CUSUM	0.0038	0.1447	6.6467	83.1364	0.2584
EWMV	0.0034	0.5702	1.7492	228.5500	0.2623

## Data Availability

Data will be available on request.

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
