# Peer review of "Method for Detecting Disorder of a Nonlinear Dynamic Plant"

_sensors, 2025, doi:10.3390/s25041256_

Round 1
Reviewer 1 Report
Comments and Suggestions for Authors
1. The accuracy of the regression model is closely associated with the original training data and the detection window. These factors should be clearly addressed in the experimental setup section.
2. The original training data are derived from simulations. It is important to discuss how the proposed model can maintain its adaptability for practical applications.
3. A comparison is made with the CUSUM algorithm, but it would be beneficial to also compare it with other intelligent algorithms.
4. For Figure 14, only the results of the CUSUM algorithm are provided. The results of the proposed method should also be included and compared.
Reviewer 2 Report
Comments and Suggestions for Authors
This article proposes CCF-AE for disorder detection in dynamic plants. It uses the cross-correlation function (CCF) between input and output signals and an autoencoder to detect deviations from normal behavior. Experiments on a pH neutralization reaction show CCF-AE outperforms the CUSUM algorithm. The authors claim CCF-AE is simpler to implement than model-based methods and better suited for nonlinear systems, with a fixed detection delay.
However, the article lacks details on parameter tuning and its impact on performance. The evaluation is limited to a single case study, and the comparison with CUSUM might be unfair given its reliance on a reference model. Furthermore, the fixed delay might be unsuitable for some applications, and the computational cost of CCF-AE is not discussed.
Furthermore, the conclusion is too brief. It should be expanded to include a summary of the experimental results, comparing the performance of CCF-AE against the CUSUM algorithm. Mentioning the fixed delay time as an advantage and suggesting future research on parameter optimization would also strengthen the conclusion.
One suggestion. The paper would benefit from a more thorough discussion of CCF-AE's advantages and disadvantages compared to other data-driven fault detection methods. While the use of CCF as a dynamic characteristic combined with an autoencoder is novel, the paper only compares CCF-AE with CUSUM, which relies on a reference model. How does CCF-AE compare to other data-driven methods like PCA, KPCA, ICA, Gaussian Mixture Models, or other machine learning approaches, especially regarding its performance on nonlinear systems, computational cost, and robustness to noise? The limited evaluation on a single pH neutralization reaction makes it difficult to assess the generalizability of CCF-AE. Further comparisons with a wider range of data-driven methods and datasets are needed to demonstrate its strengths and weaknesses.
Reviewer 3 Report
Comments and Suggestions for Authors
(1) The full name of the abbreviations should be provided in the first place of the current text.
(2) The introduction section is relaxed in logic, and it should be rephrased for improvement.
(3) The research background may be enlarged to highlight the interests of various readers from different fields.
(4) The details in the current research's experimental section are insufficient. Some experimental additions may increase the degree of matching between the current research and the Sensors journal.
(5) The title is broad for the current research scope and content.
(6) How do the authors define the parameters of the neutralization reactor's basic model and the neutralization reactor's nominal operating points?
(7) Authors presented the algorithms for detecting discord using CCF-AE (a) and the algorithm for detecting discord using CUSUM (b) in Fig. 4. In the result section, the direct results of the two algorithms are shown in 4.2 and 4.3. Still, a comparison of them may be interesting for readers.
(8) The current English styles can be further improved to meet the professional journal requirements.
Comments on the Quality of English LanguageThe current English styles can be further improved to meet the professional journal requirements.
